# Inverse Simulation for Extracting the Flow Characteristics of Artificial Snow Avalanches Based on Computation Fluid Dynamics

**Kenichi Oda \*, Katsuya Nakamura, Yoshikazu Kobayashi and Jun-ichi Suzumura**

Department of Civil Engineering, College of Science and Technology, Nihon University, Kanda-Surugadai 1-8-14, Chiyoda-ku, 101-8308 Tokyo, Japan; nakamura.katsuya@nihon-u.ac.jp (K.N.); kobayashi.yoshikazu@nihon-u.ac.jp (Y.K.); suzumura.junnichi@nihon-u.ac.jp (J.-i.S.)

\* Correspondence: oda.kenichi@nihon-u.ac.jp

**Abstract:** Many numerical analysis methods for predicting the motion of soil or snow avalanches may set the characteristics of the dynamic friction obtained from field tests, such as the measurement of real avalanche and model slope tests. However, the friction characteristics of the actual flow of objects are influenced by changes in the velocity and density of the flowing objects and are not necessarily constant. In addition, determining the shear strain rate dependence of the frictional properties at low shear strain rates is important for accurately predicting the avalanche reach distance. In this research, model tests using a rotating drum device were carried out, and an artificial snow avalanche was generated. Then, the flow velocity distribution of the flow was extracted by organizing the motion of the artificial snow avalanche flowing in the drum device using the Digital Image Correlation method. Moreover, the changing characteristics of the viscosity coefficient of the pseudo flow were estimated using an inverse simulation. For the results, it was suggested that the method of estimating flow characteristics and friction characteristics from the artificial avalanche generated by the rotating drum and the time inverse analysis proposed in this study was effective, but it is necessary to confirm the issue of the need for a similar analysis using real scale. If it is found to be applicable to real scales in the future, it will contribute to the development of this field because it will expand the range of methods for analyzing avalanches using model experiments.

**Keywords:** snow avalanche; rotating drum device; inverse simulation; viscosity coefficient; friction characteristic

## 1. Introduction

Since Voellmy proposed a numerical model for snow avalanche in 1955 [1], methods for predicting avalanche motion have been used at the field level. In addition, the introduction of simulation based on computational fluid dynamics has been realized with the development of computing equipment since 2000. Some of the available models are AVAL-1D, e.g., Christen et al. [2], MN2D, e.g., Naaim et al. [3], TITAN2D, e.g., Patra et al. [4], SAMOS, e.g., Sailer et al. [5], and RAMMS, e.g., Christen et al. [6]. These tools have been mainly used in Switzerland, Austria, and Italy and can predict the run-out distance and flow velocity of a snow avalanche in two and three-dimensional terrain. In order to reproduce snow avalanches using numerical analysis, it is important to determine the material constants of the snow that constitutes the snow avalanche. When using simulations such as RAMMS and SAMOS for wet snow avalanches that occur during the snowmelt season in Japan, it is necessary to be require special care for material constants and frictional characteristics of snow. Christen et al. [6] and Bühler et al. [7] reported a close to actual reproduction by adjusting the dynamic friction coefficient of the avalanche to simulate wet snow avalanches by using the RAMMS model. However, there is no

report that famous simulation software with a proven track record overseas is effective for avalanches in Japan.

On the other hand, development of simulations by using computational fluid dynamics for wet snow avalanches is being carried out independently in Japan. An example of snow avalanche simulation adapted to Japanese using computational fluid dynamics, which was performed in the past, is shown below. Oda et al. [8] conducted simulations using computational fluid dynamics on the runout distance and impact force of the snow avalanche flow obtained in an artificial snow avalanche experiment. Nakai et al. [9], Yamaguchi et al. [10], Oda et al. [11], and Sawada et al. [12] conducted simulations using computational fluid dynamics proposed by Oda et al. [8] obtained in an real avalanche and they reported that it was effective for consider countermeasures to the avalanche. Additionally, Titan2D, which is used as a simulation of pyroclastic flow, was applied to verify the effectiveness of snow avalanche motion by Mori et al. [13]. Titan2D, used by Mori et al. express spatial viscosity characteristics as non-linear, and these methods change the friction characteristics near the boundary between a snow avalanche and slope. In the model of Oda et al. the equivalent viscosity coefficients with spatial and temporal variations in the Bingham fluid model are introduced to represent the avalanche motion.

A change in viscosity characteristics was introduced in these methods, and the Coulomb friction law was used as a constitutive law. In these basic constitutive models, friction characteristics are introduced as static parameters. However, the friction coefficient was adjusted as a dynamic parameter depending on the density and temperature energy of the snow in order to respond to the complicated movements that match the real avalanche. The Bingham fluid model is often used to reproduce the motion of sediment flow and pyroclastic flow. However, it is difficult to use the Bingham fluid model alone to model the motion of avalanches, because the object density of the avalanche is lower than that of sediment flow and pyroclastic flow, and the drag effect in the low shear strain region may be expressed excessively, and the viscous properties of the avalanche have been reported to exhibit dilatancy properties in the high speed region (high shear strain region). The model used by Oda et al. [8] introduces equivalent viscosity coefficients, which may exhibit dilatant fluid or pseudoplastic fluid like behavior in space and time, but it cannot adequately represent dilatant fluid like behavior in the high shear strain rate region or low viscosity in the low shear rate region because of the drag being limited by the yield shear stress. In particular, the viscosity decline in the low shear strain rate region is difficult to represent with a simple non-Newtonian viscosity model because of the different behavior around the lower layers of the avalanche and between the middle and upper layers. As a method of expressing the friction coefficient that dynamically changes according to the real avalanche, a method of measuring the friction coefficient of the avalanche by observing the real avalanche [6,14] or the artificial avalanche obtained from slope model test [10–12] has been conducted. Kern et al. [15] reported that the HerschelBulkley and Cross Model combining the Bingham fluid model and the Power-Low model successfully represented the weak layer of the avalanche and reproduced the flow without the effect of changing the friction coefficient around the bottom layer. However, they report that it is difficult to perform multiple experiments under the same conditions because of the difficulty of using the avalanche chute in a field environment, and that there is a problem to deeply understand the inner structure of the avalanche. In order to express the dynamically changing friction coefficients for real avalanches, it is desirable to make adjustments based on measurements of real avalanches and large artificial avalanches. However, it is difficult to cover all avalanches because the cost of avalanche experiments and techniques depend on the results.

On the other hand, in order to understand the friction characteristics of a moving object, it is necessary to conduct an experiment that can artificially reproduce a flowing state. Ueda et al. [16] measured the shear band of sand using a small direct shear test apparatus. In these studies, a moving image of the movement of soil particles using a shear test was saved, and the displacement of the soil particles was obtained from the moving image using the image correlation method. This method can indirectly determine the frictional characteristics of an object, for example, the extraction method of

stress history using image analysis. Moreover, several experimental methods using a rotating drum device have been reported as approaches to the continuous observation of large deformations of granular materials. Pudasani et al. [17] conducted a flow measurement using a rotary drum device to elucidate the motion mechanism of avalanches of granular snow. Additionally, Hill et al. [18] reported that experiments using rotary drum devices are mainly used in fields, such as powder technology and mechanical engineering, and studies confirm the mixing and separation of materials. According to these reports, the sample in a rotary drum device has two movements, one is upward flow, and the other is downward flow. Additionally, the boundary surface between the upward flow and the downward flow draws an arc shape. In this downward flow area, the granular material always forms a steady flow from the upper side to the lower side. Therefore, it is thought that this area is optimal as a device for observing, from the side perspective, the internal situation of a snow avalanche flowing down a slope.

In this research, model tests, such as the rotating drum device based on the above method, are carried out, and an artificial snow avalanche such as stream type is generated. Then, the flow velocity distribution of the flow is extracted by organizing the motion of the artificial snow avalanche flowing in the drum device using the Digital Image Correlation method. Moreover, the changes in the viscous properties in the pseudo-flow were estimated by time inverse analysis using the numerical fluid model proposed by Oda et al. It should be noted that the phenomena that can be treated at the scale of a rotating drum device are not large enough to reproduce the real avalanche flowing at high speed. Therefore, the phenomena discussed in this study are limited to viscous properties in the low shear strain rate region, which can affect the runout distance of the avalanche. In addition, to estimate the viscosity characteristics using the proposed method in this study, the constitutive law is solved from the previous numerical models, which are discretized using the finite difference method, and the temporal variation of the advection term and non-advection term. In order to discuss the basic knowledge, this study was conducted under the conditions that artificial avalanches using a rotating drum device were unaffected by temperature and external forces during the experiment, and dry snow samples were used. Therefore in this report, the viscous properties of a pseudo-flow of dry snow in a rotating drum as a fluid were confirmed.

## 2. Rotating Drum Experiment

### 2.1. Experimental Conditions

In this experiment, an experimental sample was placed in a rotary drum device and rotated. The rotary drum device had an inner diameter of 118 mm and a length of 130 mm. Figure 1 shows the rotary drum device used in these experiments. In this study, we calculated the friction coefficient by observing an arc shape flowing above the boundary surface.

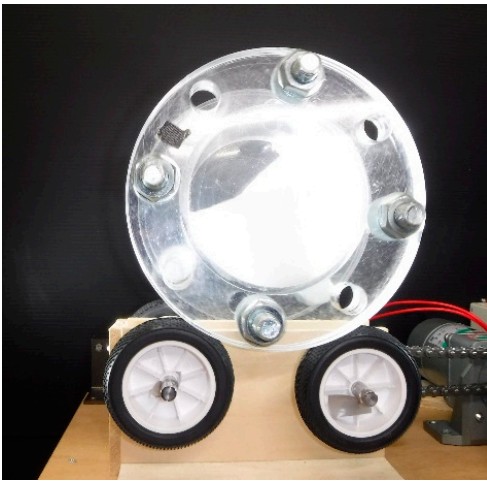

**Figure 1.** Rotating drum used in this study. An acrylic cylindrical container is set on a turntable.

Laboratory experiments were conducted using snow kept in a room at −20 degrees. This room was made available by the Research Institute for Natural Hazards and Disaster Recovery, Niigata University. Table 1 lists the snow classifications, e.g., Izumi [18]. All the snow used in this experiment was classified as 'Granular snow,' and the experiments were carried out in the same low-temperature room. In the experiments, snow that had been adjusted in advance was used. Figure 2 shows a particle image of the adjusted snow. In these experiments, three snow particle sizes (1 mm, 2 mm, and 4 mm) and three rotating speeds (10 rpm, 15 rpm, and 20 rpm) were used. Snow particles were created from blocks that had contained the natural snow in the low-temperature room. The procedure for generating snow particles from a block was as follows.

**Table 1.** Classification of snow quality according to the density of the snow [19].

| Classification | Density $\rho$ [kg·m$^{-3}$] |
| --- | --- |
| Fresh snow | 50.1~150.1 |
| Lightly compacted snow | 50.1~500.0 |
| Compacted snow | 250.1~500.0 |
| Granular snow | 300.0~500.0 |
| Lightly granular snow | About 300.0 |
| Frost granular snow | About 300.0 |

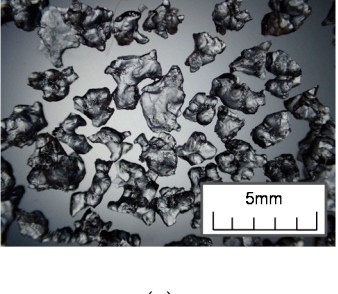
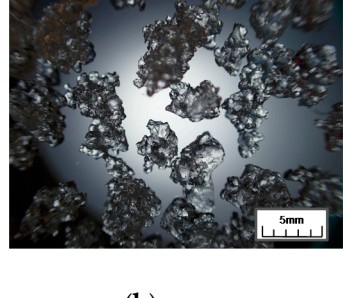
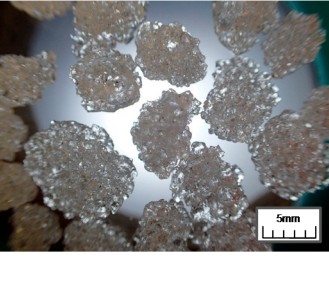

(**a**)　　　　　　　　　　(**b**)　　　　　　　　　　(**c**)

**Figure 2.** Particle image of the adjusted snow: (**a**) particle size is 1 mm; (**b**) particle size is 2 mm; (**c**) particle size is 4 mm.

- Cut the snow block into 2–3 cm blocks
- Remove the ice part, where the melted water freezes
- Crush the snow blocks using a crusher machine with a 6 mm diameter mesh
- Screen the milled snow and classify it according to its particle size (4 mm, 2 mm, or 1 mm)

Figure 3 shows the particle size distribution of the snow particles used in this experiment. The snow particles generated in this experiment had an average particle size of 0.1 mm or less. However, considering that the relationship between snow particles and adhesion strength is inversely proportional [20], the experimental samples with a particle size of 1 mm or more were used. Figure 4 shows photos of the procedure for creating snow particles.

For each experiment, three trials were conducted to verify reproducibility. The rotational speed of the rotating drum device was set to three patterns: 10, 15, and 20 rpm. The internal volume of each sample was set to about 3/4 of the volume of the drum. Table 2 shows the experimental conditions of this experiment. A high-speed camera (K-III: KATO KOKEN CO, LTD, Tokyo, Japan) was used for flow observation in a rotary drum device. The picture resolution was set at 1280 × 1024 pixels, the frame rate was set at 1000 fps, and the observation was performed for 1.6 seconds (total of 1636 frames).

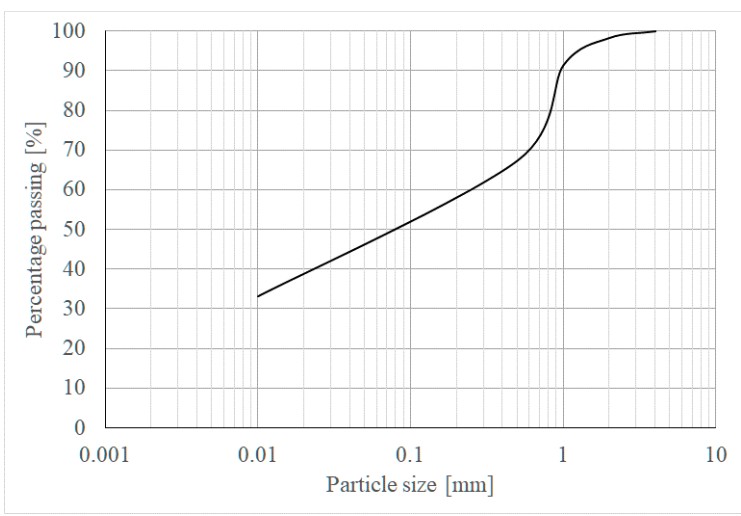

**Figure 3.** Particle size distribution of the snow particles used in this experiment.

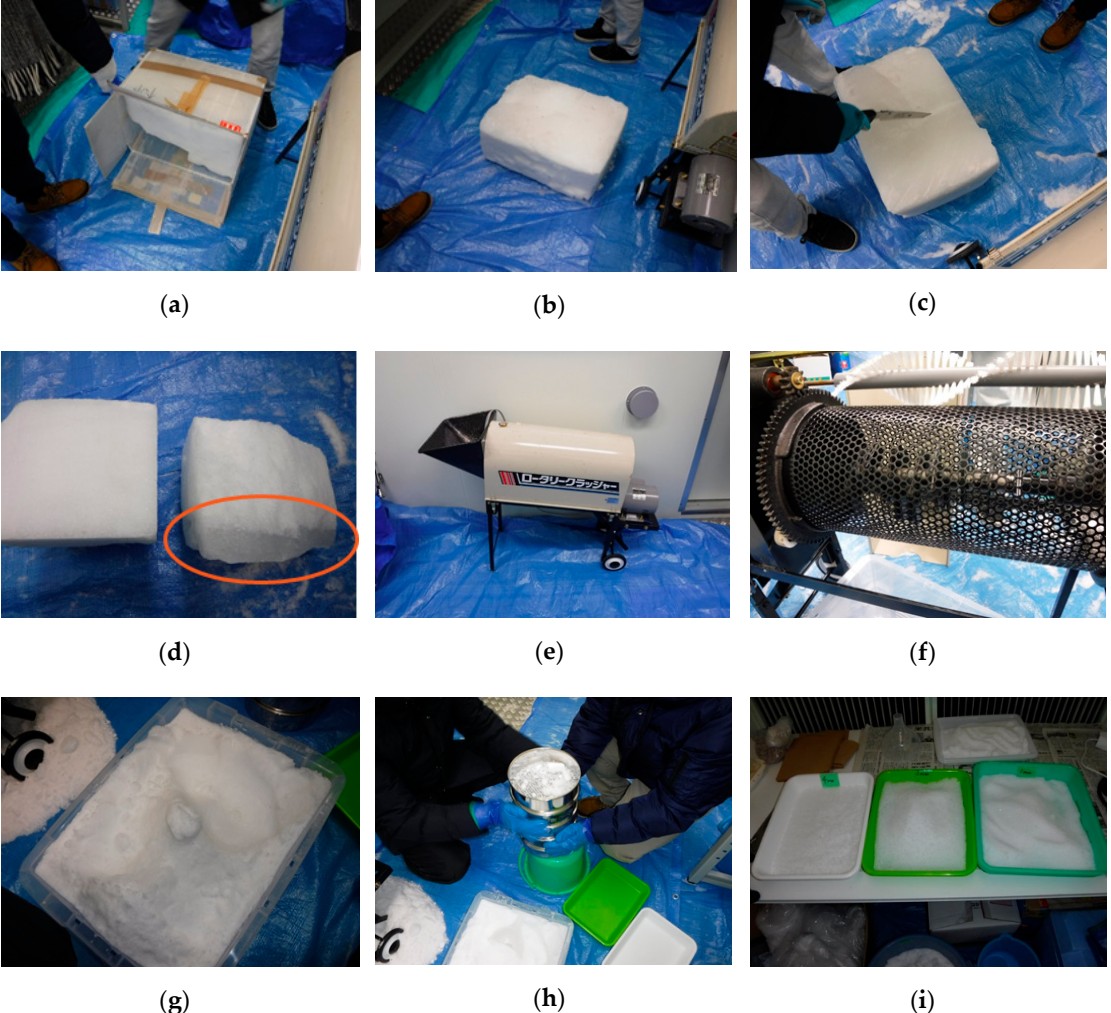

**Figure 4.** Photos of the procedure for creating snow particles: (**a**) taking the snow chunk from the box; (**b**) the snow chunk; (**c**) cutting the snow chunk into some pieces; (**d**) removing the ice conditions; (**e**) crushing snow with a crusher machine; (**f**) internal structure of the crusher machine with a 6 mm diameter mesh; (**g**) milled snow; (**h**) screening the milled snow; (**i**) classifying the snow according to its particle size (4 mm, 2 mm, or 1 mm).

**Table 2.** Experimental conditions.

| Case | Average of Snow Particle [mm] | Rotational Speed [rpm] | Apparent Density [kg·m$^{-3}$] | Number of Times |
|------|-------------------------------|------------------------|--------------------------------|-----------------|
| H-1 | 1 | 20 | 363 | 3 |
| H-2 | 2 | 20 | 335 | 3 |
| H-4 | 4 | 20 | 308 | 3 |
| M-1 | 1 | 15 | 363 | 3 |
| M-2 | 2 | 15 | 335 | 3 |
| M-4 | 4 | 15 | 308 | 3 |
| L-1 | 1 | 10 | 363 | 3 |
| L-2 | 2 | 10 | 335 | 3 |
| L-4 | 4 | 10 | 308 | 3 |

*2.2. Extraction of Flow Velocity Distribution*

In this study, particle motion was obtained from the image of the flow using particle image velocimetry (PIV), and the arc shape part was extracted. In the image analysis by PIV, the motion of snow particles was emphasized by applying a Kirsch filter of a differential filter that emphasizes the edge of the image as preprocessing, e.g., Wang [21]. Figure 5 shows a picture of the Kirsch filter image.

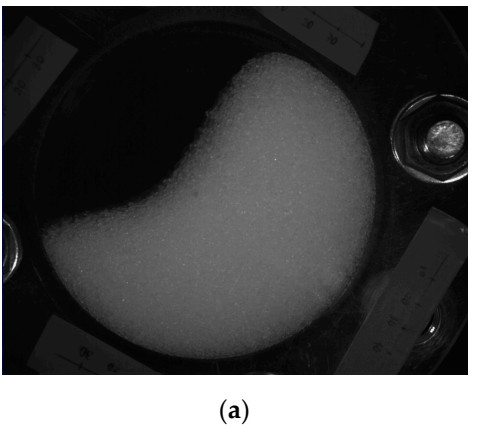
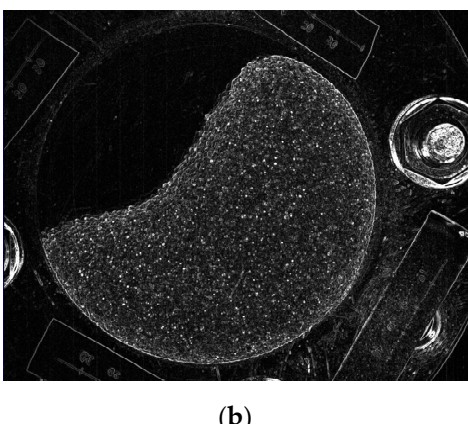

(**a**)  (**b**)

**Figure 5.** Picture of the Kirsch filter image: (**a**) a photo of the experiment taken with a high-speed camera; (**b**) applying the Kirsch filter.

Besides, the examination area of the particle was $10 \times 10$ pixels, the search area was $32 \times 32$ pixels, and the calculation grid width was 1.0 mm (an equally spaced grid in all cases). Moreover, the calculation mesh was set to be inclined by 35 degrees counterclockwise to make the flowing direction parallel to the horizontal axis.

## 3. Method of Calculating the Friction Coefficient

Many rheological models have been proposed to describe the behavior of flowing sediment. Oda et al. [8] used a Bingham fluid model with equivalent viscosity coefficients [22] as a viscosity model to reproduce avalanche flow in wet snow. In a simple shear state, the Bingham model can be described as a linear expression between the shear stress and the shear strain rate as follows:

$$\tau = \eta_0 \dot{\gamma} + \tau_y, \tag{1}$$

where $\tau$ is the shear stress, $\eta_0$ is the viscosity after yield, $\dot{\gamma}$ is the shear strain rate, and $\tau_y$ is the yield shear strength. In order to describe both the cohesive and frictional behavior of the material, Coulomb's failure criterion is introduced as the yield shear strength into the Bingham model. The yield criterion is defined by the following equation:

$$\tau = \eta_0 \dot{\gamma} + c + \sigma_n \tan \phi, \tag{2}$$

where $c$ is the cohesion, $\phi$ is the angle of internal friction, and $\sigma_n$ is the normal stress. Since the snow avalanche is assumed to be fluid, the normal stress can be replaced by the hydrostatic pressure $p$, as shown in the following equation:

$$\tau = \eta_0 \dot{\gamma} + c + p \tan \phi. \tag{3}$$

An equivalent viscosity $\eta'$ can be obtained from the above equation as

$$\eta' = \frac{\tau}{\dot{\gamma}} = \eta_0 + \frac{(c + p \tan \phi)}{\dot{\gamma}}. \tag{4}$$

By introducing an equivalent viscosity coefficient, the viscoplastic model of the fluid represented is not the general Bingham model, but a viscoplastic model that exhibits varying behavior in time and space. The above equivalent viscosity is used to consider the effect of the evolving shear strain rate on the flow behavior of the material. In two- and three-dimensional stress states, the equivalent viscosity can be generalized as

$$\eta' = \eta_0 + \frac{(c + p \tan \phi)}{\sqrt{2 V_{ij} V_{ij}}}. \tag{5}$$

In which

$$V_{ij} = \frac{1}{2}\left(\frac{\partial u_i}{\partial x_j} + \frac{\partial u_j}{\partial x_i}\right), \tag{6}$$

where $u_i$ is the velocity vector. The snow is a substance that is compressed by temperature changes and external forces. However, when avalanche motion is treated fluidly, the compressibility of the flow is often treated as low because of the relationship between the velocity of the flow and the speed of sound. One of the reasons for this is because it is difficult to account for complex phenomena such as density changes caused by energy dissipation in the case of pronounced surging inside an real avalanche. In this study, snow avalanche is assumed to be an incompressible fluid. Therefore, the stress can be expressed as follows:

$$\sigma_{ij} = -p\delta_{ij} + 2\eta' V_{ij} = -p\delta_{ij} + \eta'\left(\frac{\partial u_i}{\partial x_j} + \frac{\partial u_j}{\partial x_i}\right). \tag{7}$$

The following equations are used as governing equations:

$$\frac{\partial u_i}{\partial t} + u_j \frac{\partial u_i}{\partial x_j} = \frac{1}{\rho}\frac{\partial \sigma_{ij}}{\partial x_i} + g_i, \tag{8}$$

$$\frac{\partial u_i}{\partial x_i} = 0, \tag{9}$$

where $\rho$ is the total mass density of the snow, and $g_i$ is the gravity acceleration vector. Equation (8) is the linear momentum conservation law. Equation (9) is the equation of continuity. By substituting Equation (7) with Equations (8), Equation (10) can be derived:

$$\frac{\partial u_i}{\partial t} + u_j \frac{\partial u_i}{\partial x_j} = -\frac{1}{\rho}\frac{\partial p}{\partial x_i} + \frac{1}{\rho}\frac{\partial}{\partial x_j}\left(\eta' \frac{\partial u_i}{\partial x_j}\right) + g_i. \tag{10}$$

In the case of a Newtonian fluid, the viscosity coefficient is constant, and its spatial derivative is zero. However, the equivalent viscosity $\eta'$ has a spatial derivative. Therefore, Equation (8) can be used to incorporate the spatial derivative of $\eta'$. In the existing method, Equation (8) is divided into the advection term and the non-advection term to solve the fluid flow velocity that has changed over time. When Expression (10) is expressed by the advection term and the non-advection term, it is as follows:

$$\frac{\partial u_i}{\partial t} + u_j \frac{\partial u_i}{\partial x_j} = 0, \tag{11}$$

$$\frac{\partial u_i}{\partial t} = -\frac{1}{\rho}\frac{\partial p}{\partial x_i} + \frac{1}{\rho}\frac{\partial}{\partial x_j}\left(\eta' \frac{\partial u_i}{\partial x_j}\right) + g_i. \tag{12}$$

Here, Equation (11) is the advection term, and Equation (12) is the non-advection term.

In the existing method, the advection term and non-convection term were time-discretized using the SMAC method, e.g., Amsden [23]. Then, the velocity after time evolution was calculated. In this study, the equivalent viscosity coefficient and pressure are determined as unknowns. However, the velocity information can be obtained from an experiment in our research. Therefore, in order to perform an inverse simulation, the calculation procedure is performed by giving a one-step flow rate, as a known quantity, back and forth.

In this research, we attempted to calculate the internal friction angle $\phi$ shown in Equation (5), using flow velocity data, obtained by image analysis, as input parameters. The calculation step of the inverse simulation is as follows:

1. Solve the advection term using the velocity after one step $u^{n+1}$ to find the intermediate velocity $u^1$
2. Derive the Poisson equation from the pressure term of the non-advection term. Then, calculate the pressure $p$ using the intermediate velocity $u^1$
3. Solve the pressure term of the non-advection term using pressure $p$, and calculate the intermediate velocity $u^2$
4. Calculate the intermediate velocity $u^3$ from the external force term of the non-advection term using the velocity of one step before $u^n$
5. Solve the viscosity term using the intermediate velocities $u^2$ and $u^3$ to find the equivalent viscosity coefficient

To calculate the time evolution using these methods, the above procedure should be repeated. The non-advection term uses a discretized formula as follows:

$$\frac{u_i^3 - u_i^n}{\Delta t} = g_i, \tag{13}$$

$$\frac{u_i^2 - u_i^3}{\Delta t} = \frac{1}{\rho}\frac{\partial}{\partial x_j}\left(\eta' \frac{\partial u_i}{\partial x_j}\right), \tag{14}$$

$$\frac{u_i^1 - u_i^2}{\Delta t} = -\frac{1}{\rho}\frac{\partial p}{\partial x_i}. \tag{15}$$

Here, Equation (13) is the external force term. Equation (14) is the viscosity term. Equation (15) is the pressure term.

In this case, it is known that the equivalent viscosity coefficient is a parameter that varies greatly in space and time, depending on the shear strain rate. Therefore, the implicit method is applied to the time integral of the viscosity term. Here, the Poisson equation can be obtained from the pressure term using spatial differentiation and the incompressible conditional Equation (9).

$$\frac{u_i^2}{\partial x_i} = 0, \tag{16}$$

$$\frac{\partial}{\partial x_i}\left(\frac{1}{\rho}^n \frac{\partial p}{\partial x_i}\right) = -\frac{1}{\Delta t}\frac{u_i^1}{\partial x_i}. \tag{17}$$

Here, Equation (16) is an incompressible conditional equation, and the Equation (17) is the Poisson equation.

Finally, the internal friction angle is calculated from Equation (4), with the obtained viscosity coefficient and pressure. The internal friction angle can be obtained as follows:

$$\phi = \tan^{-1}\left\{\frac{\left(\eta' - \eta_0 - \frac{c}{\dot{\gamma}}\right)\dot{\gamma}}{p}\right\}. \tag{18}$$

In this study, while the snow samples can be considered to have been in a dry condition, the cohesion *c* was set to zero. When the cohesion *c* is deleted, Equations (3) and (18) can be expressed as follows:

$$\phi = \tan^{-1}\left\{\frac{(\eta' - \eta_0)\dot{\gamma}}{p}\right\}, \tag{19}$$

$$\tau = \eta_0\dot{\gamma} + p\tan\phi. \tag{20}$$

In this simulation, the viscosity after yield $10^{-10}$ Pa·s was used. From the scale of the pseudo-flow and the temperature environment of the model experiment, the pseudo-flow is assumed to be an incompressible fluid when treated as a fluid in this research. Therefore, it is important to note that the viscosity properties obtained here are not equivalent to the viscosity properties that would be extracted if these were actually measured by a viscometer, etc.

In this method, the Confined Interpolation Profile (CIP) method, e.g., Yabe [24] was applied to the calculation of the advection term related to velocity. For other terms, spatial discretization using the Finite Difference Method (FDM) was performed. In addition, the Poisson equation obtained from the viscosity and pressure terms needs to solve simultaneous linear equations. In this study, the calculation was performed using the stabilized biconjugate gradient method (Bi-CGSTAB method), e.g., van der Vorst [25]. In this case, LU decomposition is applied to the coefficient matrix to speed up the calculation.

## 4. Experimental Result

In the experiment using the rotating drum device, we focus on the downward flow part. The target area is shown in Figure 6. In this study, the flow velocity distribution is extracted using PIV in this area. The extracted flow velocity has two components, a horizontal direction and vertical direction (x direction and y direction). Figure 7 shows the vertical flow velocity distribution, and Figure 8 shows the horizontal flow velocity distribution. The flow velocity distribution shown in Figures 8 and 9 are the first frames of 1635 frames. It can be seen that the flow velocity distribution in the vertical direction does not change uniformly in any case. On the other hand, it can be confirmed that the flow velocity distribution in the horizontal direction changes uniformly. Therefore, the downward flow part is extracted based on the horizontal flow velocity distribution.

Also, in the horizontal flow velocity distribution, it can be seen that it is in a different from its form in the downward flow part. It is assumed that this is because the mesh size used in PIV is too small, compared to the average snow particle size from the experiment. In order to reduce such observation errors, the Gaussian distribution was obtained from the data of 1636 frames per observation. The error was determined using $2\sigma$, and data with a 95% confidence interval were used. In this study, the data on flow velocity were obtained and divided into 2 groups of 818 frames. Here, $u^n$ and $u^{n+1}$ are defined as the average of flow velocity in the 2 groups.

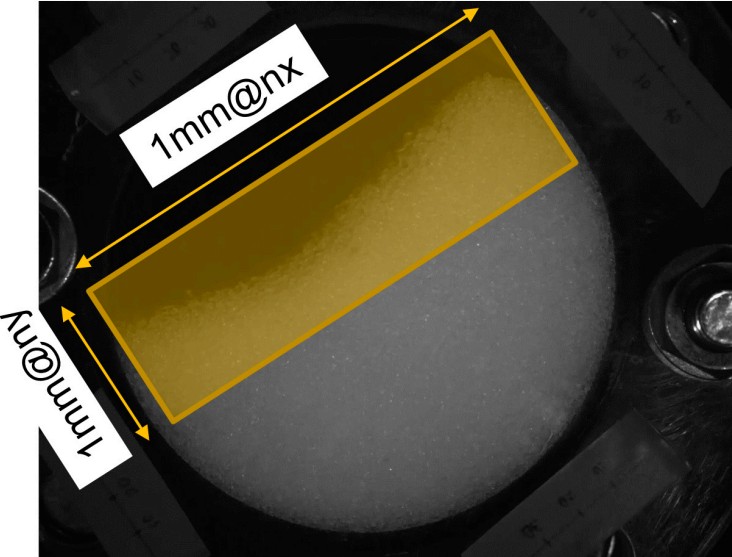

**Figure 6.** Target area for simulation.

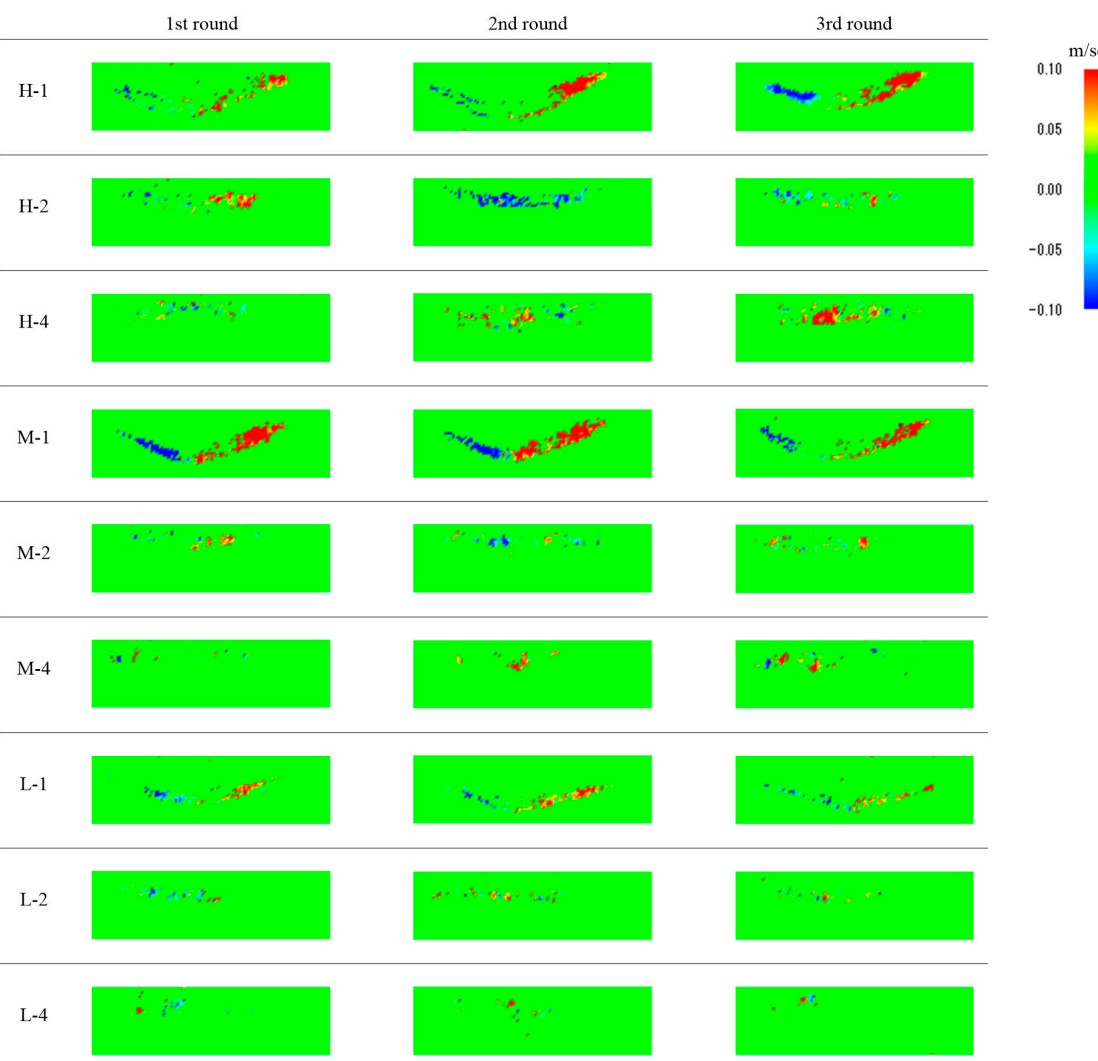

**Figure 7.** Flow velocity distributions from the results of H-1.

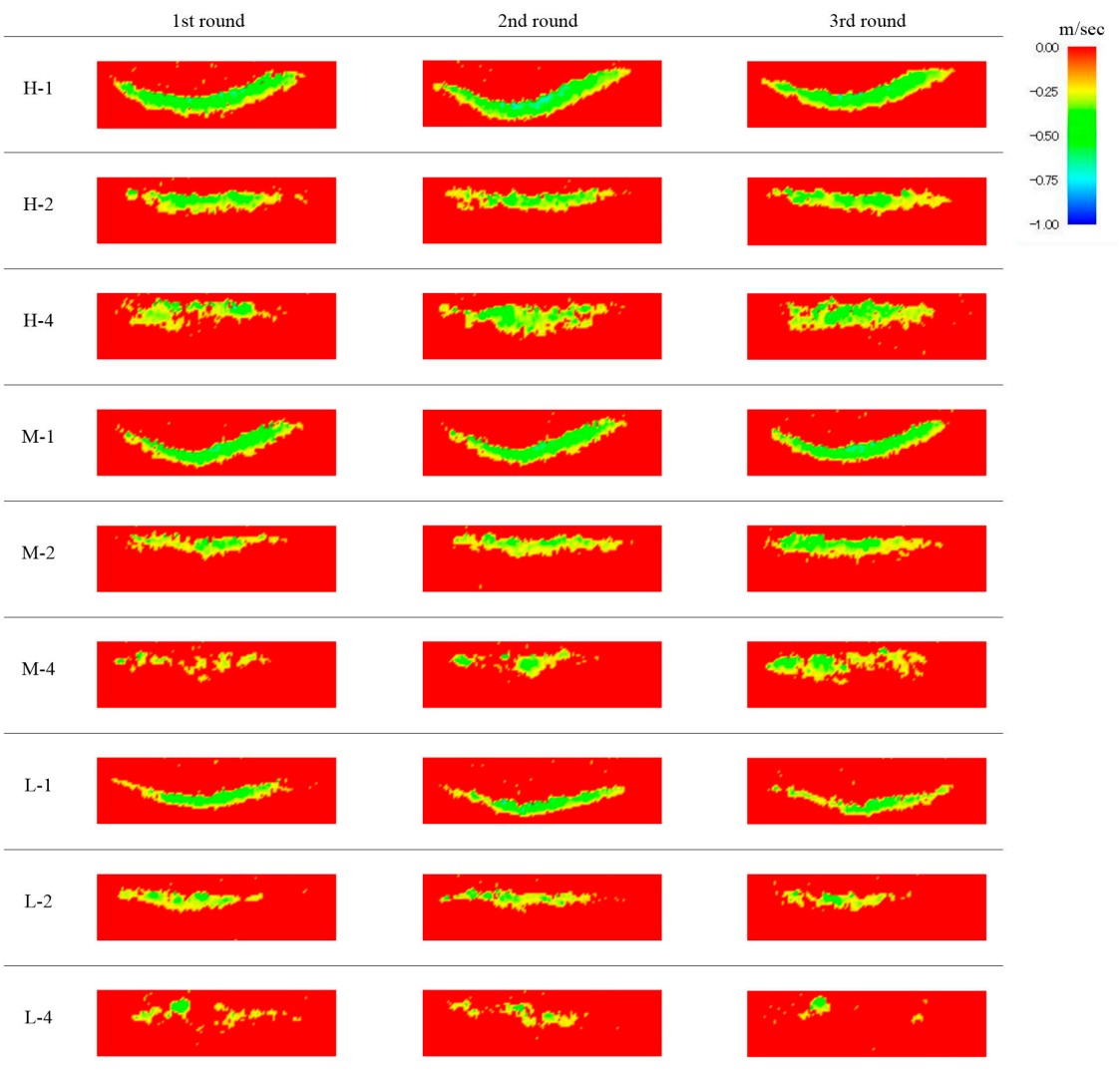

**Figure 8.** Flow velocity distributions from the results of M-1.

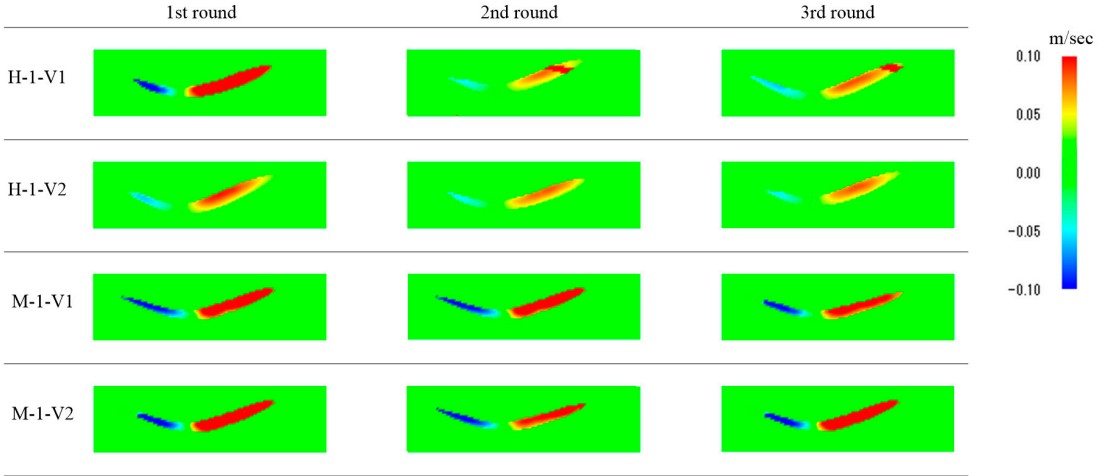

**Figure 9.** Average of the vertical flow velocity distributions: 1 is $v^n$, and 2 is $v^{n+1}$.

As a result of these processes, we obtained a relatively good flow velocity distribution from H-1 (the rotation speed is 20 rpm, and the average snow particle size is 1 mm) and M-1 (the rotation

speed is 15 rpm, and the average snow particle size 1 mm). Figures 9 and 10 show the flow velocity distributions from the results in cases H-1 and M-1. In order to analyze the movement of particles in the pseudo-flow, the flow velocities were calculated by combining the horizontal and vertical flow velocities for every 409 frames (0.409 s) of four 1636 frames, and the results of the averaged flow velocities by dividing the cross-section of the pseudo-flow in each horizontal (x-direction) direction are shown in Figures 11 and 12. The results in Figures 11 and 12 confirm that the movement of the snow particles repeatedly accelerates and decelerates, even though these particles are in the same position. This suggests that the movement of snow particles in the pseudo-flow is a reflection of the roll-wave-like movement discussed in the reports of Vriend et al. [26] and Köhler et al. [27]. The change in flow velocity is greatest around the middle of the pseudo-flow and is dependent on the magnitude of the velocity. In other words, the surge in the pseudo-flow may have affected friction by facilitating the movement of snow particles.

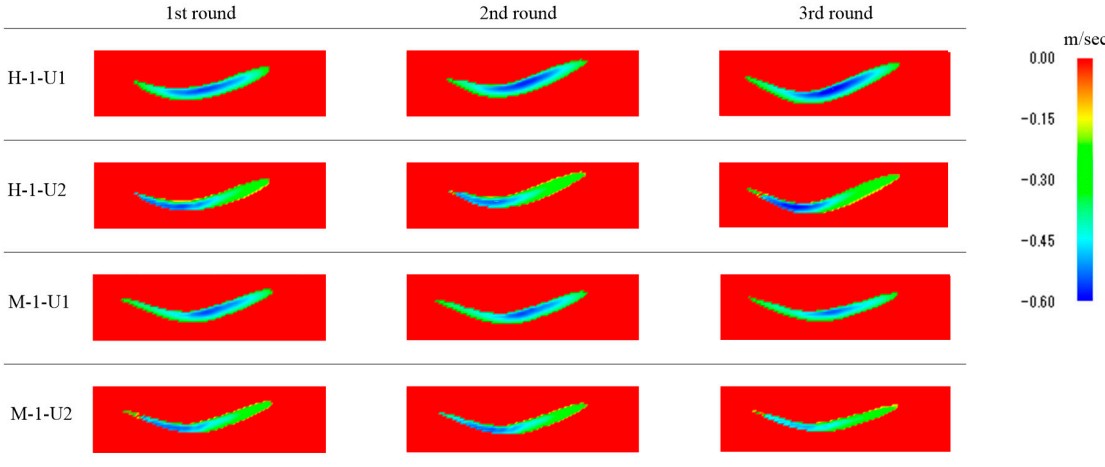

**Figure 10.** Average horizontal flow velocity distribution. The flow velocity is expressed as negative because the right direction is positive: 1 is $u^n$, and 2 is $u^{n+1}$.

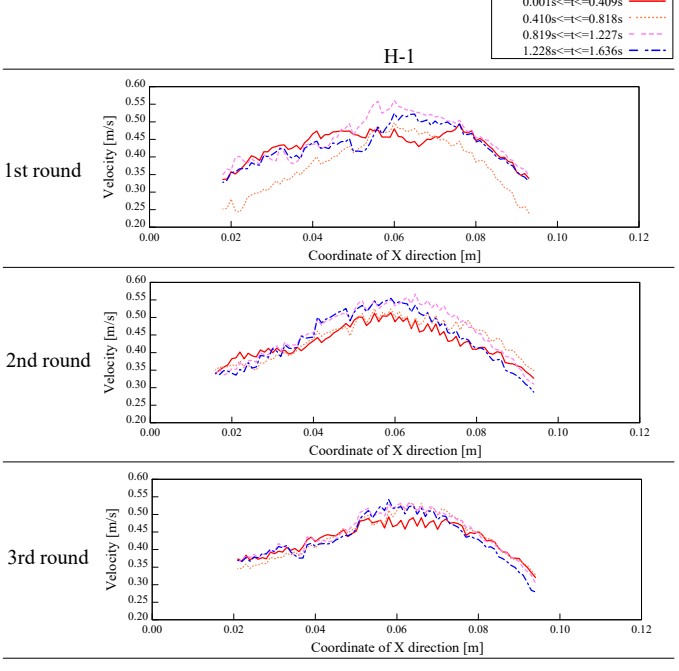

**Figure 11.** Relationship of the flow velocities in each cross section by combining the horizontal and vertical flow velocities for every 409 frames (0.409 s) of four 1636 frames from H-1.

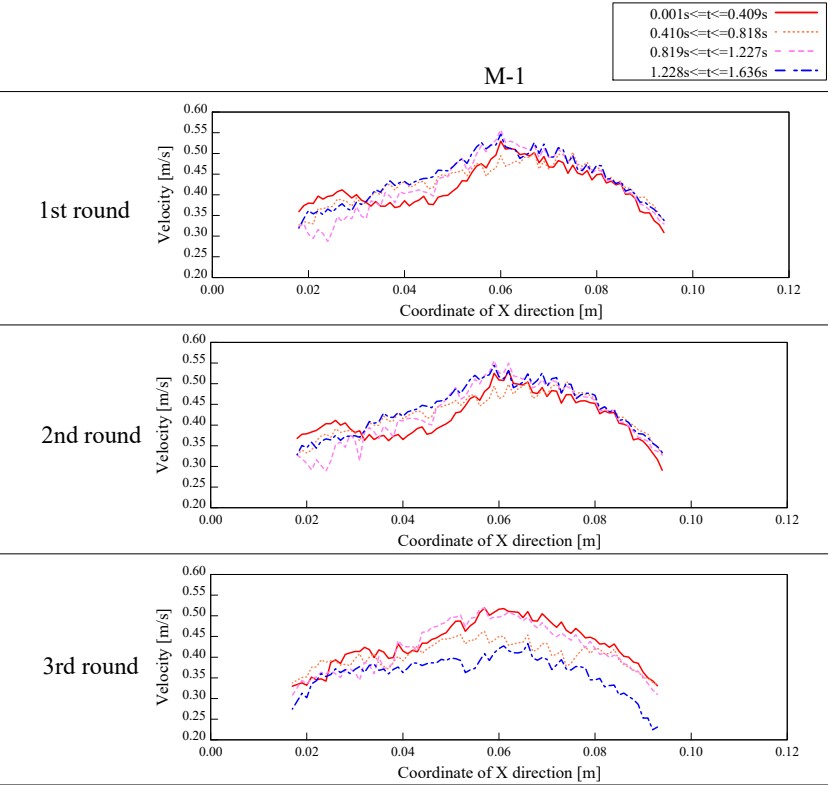

**Figure 12.** Relationship of the flow velocities in each cross section by combining the horizontal and vertical flow velocities for every 409 frames (0.409 s) of four 1636 frames from M-1.

The Froude number in the pseudo-flow is then estimated in order to understand the flow conditions of the pseudo-flow. When the layer thickness of the pseudo-flow is the characteristic length and the average flow velocity per layer thickness is the characteristic velocity, the Froude number in the pseudo-flow is as follows,

$$F_r = \frac{U_m}{\sqrt{gL_m}}, \tag{21}$$

where $U_m$ is the characteristic velocity, $L_m$ is the characteristic length, and $g$ is the gravitational acceleration. The results of the Froude number obtained by Equation (21) are shown in Figures 13 and 14. The Froude number is sharply increased at both ends of the pseudo-flow (near the beginning and end of the pseudo-flow), but at other locations, it is found to be around 1.5 to 2.0. This suggests that the pseudo-flow is in the supercritical fluid. On the other hand, the rapid decrease in the thickness of the pseudo-flow is considered to be the reason for the large change in the Froude number at both ends. Since both ends of the pseudo-flow are in the boundary layer where the flow changes, the motion may be different from the rest of the area. Assuming that the density and gravitational field of the pseudo-flow and the real avalanche are similar, and that the characteristic length of the real avalanche is 0.5 m of avalanche thickness, the characteristic flow velocity of the real avalanche is obtained as follows,

$$U = \frac{U_m}{\sqrt{L_m/L}} \approx 3.0\frac{m}{s}. \tag{22}$$

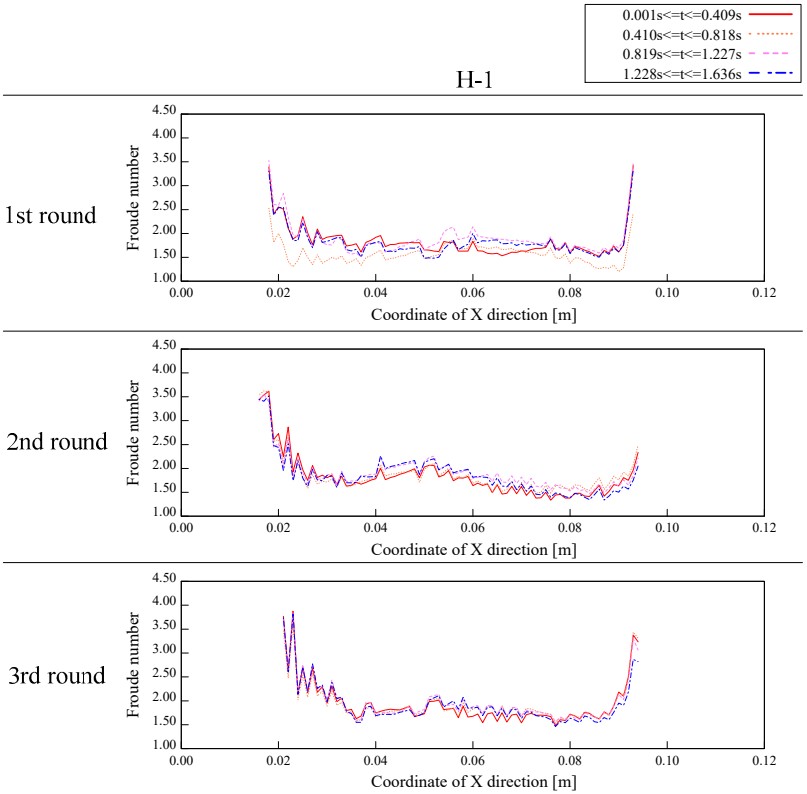

**Figure 13.** Relationship of Froude numbers in each cross section from H-1.

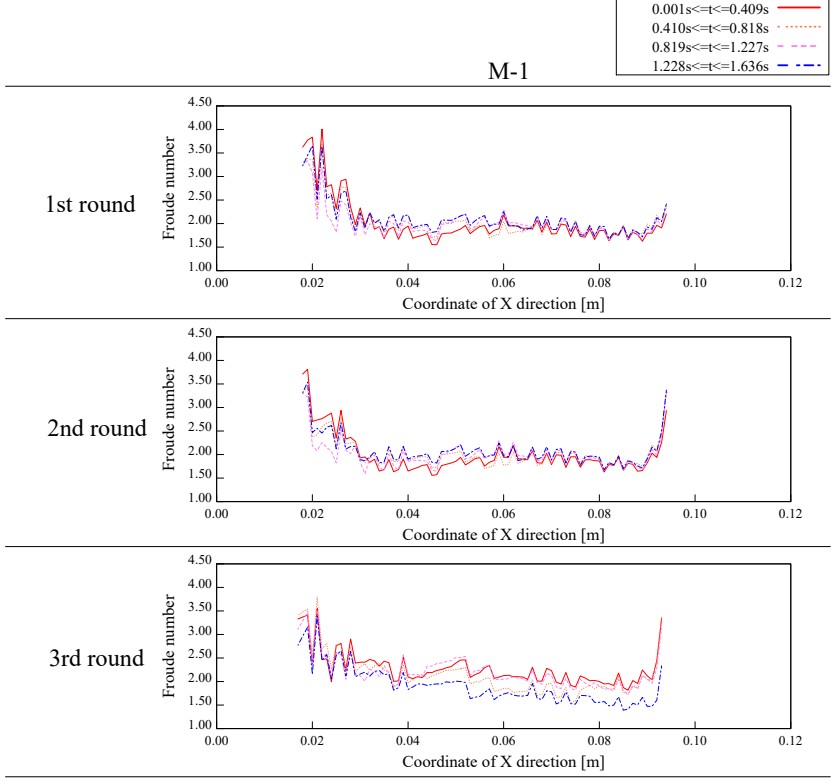

**Figure 14.** Relationship of Froude numbers in each cross section from M-1.

In addition, the pseudo-flow thickness is about 0.5 cm and the flow length is 12.5 cm in diameter of the rotating drum, it is estimated that a thickness of 50.0 cm corresponds to a flow length of about 12.0 m. The Froude number obtained here is defined as the average flow velocity for each thickness of the pseudo-flow and the dimensionless number at that position. Naturally, this is only if the dominant force is assumed to be the two forces of inertia and gravity. Therefore, the dominant force cannot be genuinely compared to a real avalanche, where the viscous force cannot be ignored. In this case, the Reynolds number for non-Newtonian viscosities is used for comparison with the real avalanche. Kern et al. [15] defined the Reynolds number for a non-Newtonian fluid focusing on the weak layer of the artificial avalanche flowing over the avalanche chute as follows.

$$R_e = \frac{\rho H v(h)}{\tau / [v(h)/h]} = \frac{\rho H v(h)^2}{\rho g h H \sin \theta} = \frac{v(h)^2}{g h \sin \theta}, \tag{23}$$

where $H$ is the thickness of the layer, $h$ is the depth from the slope to the weak layer, $v(h)$ is the flow velocity in the weak layer, $\tau$ is the shear stress at the bottom of the slope, $\rho$ is the density, and $\theta$ is the inclination angle of the slope. In this way, this study examines what kind of flow occurs around the weak layer inside the avalanche that flows over the avalanche chute. In the pseudo-flow of this study, the structure around the bottom layer may be more complicated than that of the avalanche chute, but the pseudo-flow is considered as fluid and evaluated in the same way. The coefficients used here are $v(h)$, which is the velocity of the pseudo-flow shown in Figures 11 and 12 averaged over all 1636 frames; $h$, which is the depth of the pseudo-flow with half the thickness of the layer; and $\theta$, which is the angle between the two ends of the pseudo-flow and the line connecting the two ends of the pseudo-flow. The Reynolds numbers obtained in this experiment are shown in Figures 15 and 16. These results show that the Reynolds number at both ends of the pseudo-flow is high, as well as the Froude number, but the Reynolds number is around 20. The Reynolds number obtained here is used to calculate the real avalanche velocity estimated by the fluid number, and $U = U_m h / h_m \approx 3.0 \, [m/s]$ is obtained, which is similar to the Froude number. From these results, it is assumed that the pseudo-flow obtained in the experiment corresponds to laminar and supercritical fluid.

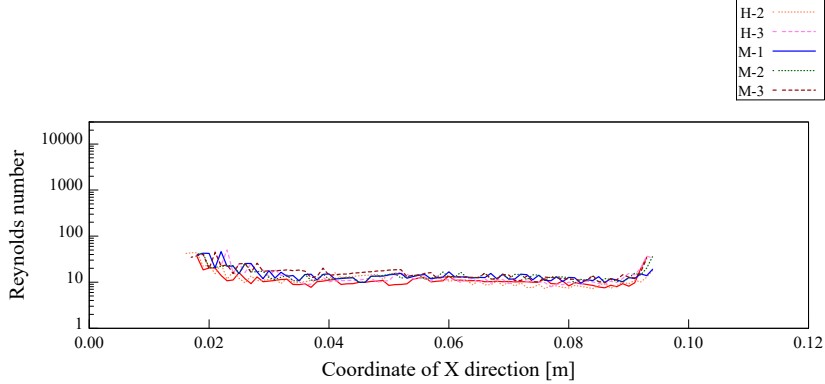

**Figure 15.** Relationship of Reynolds numbers in each cross section.

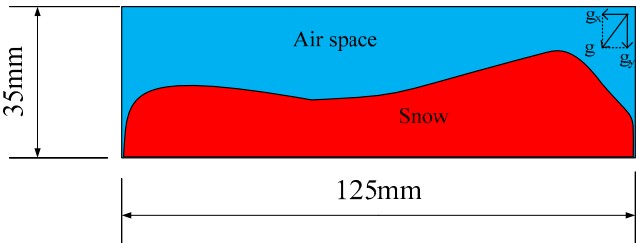

**Figure 16.** Numerical model used in the simulation.

Therefore, in this research, we formulate our discussion on the basis of the velocity distribution of H-1 and M-1.

## 5. Inverse Simulation Using the Experimental Results

### 5.1. Conditions of Numerical Analysis

In this simulation, a two-dimensional numerical model was used, based on the results of experiments using a rotary drum device. Figure 16 shows the numerical model used in the simulation. The snow density was based on the results of the experiments and were set at 363 (1 mm), 335 (2 mm), and 308 (4 mm) kg/m$^3$. The airflow was also incorporated, and the air density and viscosity coefficient were set at 1.25 kg/m$^3$ and $2.00 \times 10^{-5}$ Pa·s, respectively. Table 3 shows the input parameters of the snow. A uniform cartesian mesh was applied in the simulation, using a 1 mm mesh size ($\Delta x = \Delta y = 1$ mm). The calculation domain was inclined to minimize the calculation costs, and the incline of the model slope was specified in the form of a horizontal component of gravity (35 degrees counterclockwise).

**Table 3.** Input parameters of snow.

| Particle Size [mm] | Density $\rho$ [kg·m$^{-3}$] |
|---|---|
| 1.00 | 363 |
| 2.00 | 335 |
| 4.00 | 308 |

In this study, we performed two-dimensional inverse simulation using the flow velocity data of the artificial flow, obtained from the rotating drum test. The flow velocity data targeted an area determined as a downstream part in the flow velocity distribution of the horizontal component, obtained from the PIV. Therefore, in the inverse simulation, the part determined to be the downstream part between the two steps is processed as the fluid part, and the other area is treated as the boundary part.

### 5.2. Distribution of Mechanical Parameters

The simulation results for H-1 and M-1 are as follows. Figure 17 shows the pressure distribution, Figure 18 shows the distribution of the viscosity coefficient, and Figure 19 shows the distribution of the internal friction angle. In addition, a simulation was performed on the results from the experiments repeated three times. The flowing direction is from left to right.

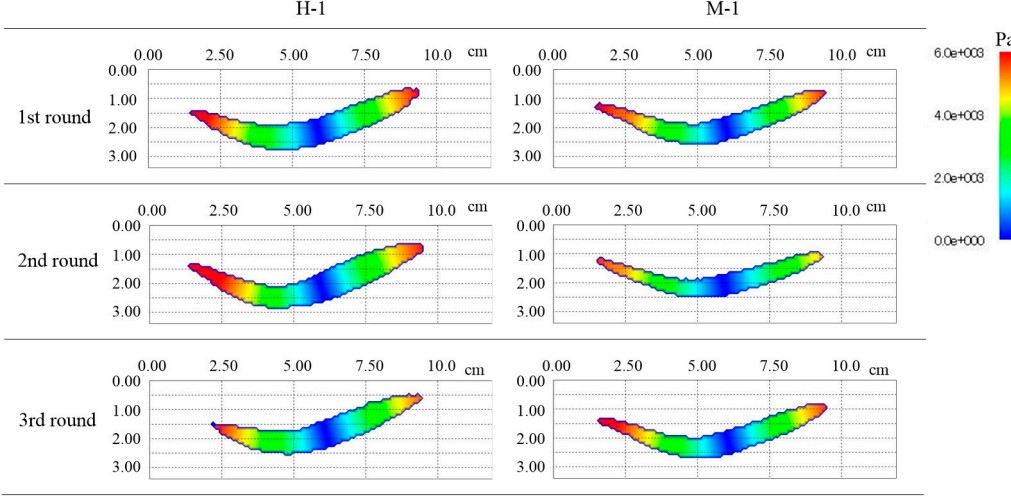

**Figure 17.** Pressure distribution from the simulation results.

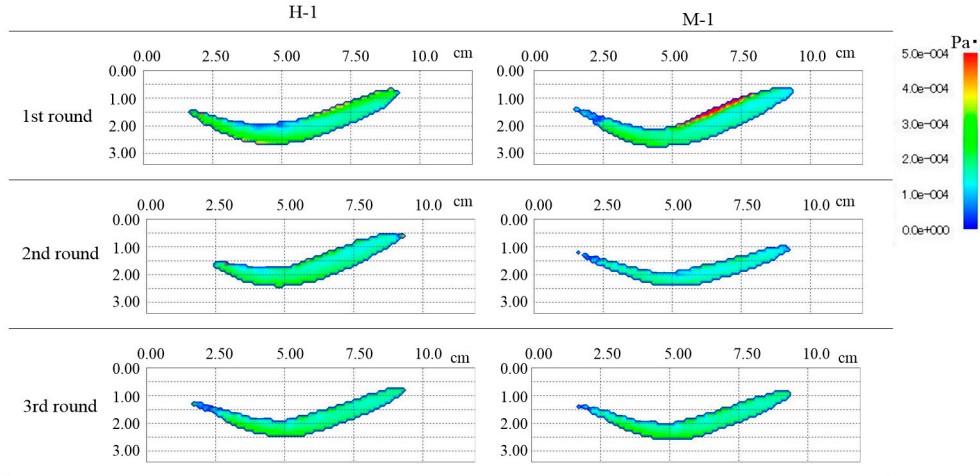

**Figure 18.** Distribution of the viscosity coefficient from the simulation results.

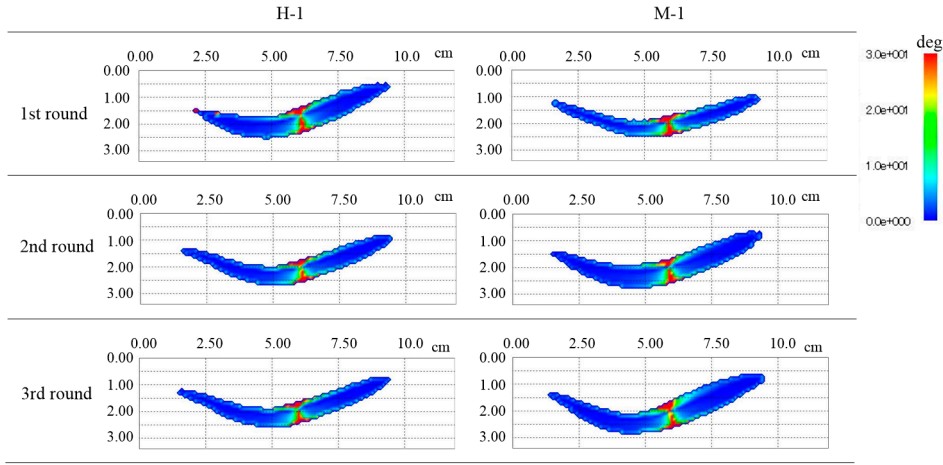

**Figure 19.** Distribution of the internal friction angle from the simulation results.

From the pressure distribution, it can be seen that the left and right edge of H-1 and M-1 have a high value. Additionally, it is shown that the center of the downstream part is the minimum value. On the other hand, it can be seen that the value of the viscosity coefficient changes largely at the boundary of the downstream part, and the other part is distributed around $2.00 \times 10^{-4}$ Pa·s. Furthermore, it can be seen that the internal friction angle is a large value at around the center of the downstream part, where the pressure value is lower. Therefore, it can be found that the internal friction angle is a changing parameter, depending on the flow location.

*5.3. Discussion of the Simulation Results*

In order to discuss the mechanical parameters, obtained by the simulation, in more detail, the viscosity coefficient and flow velocities used in simulations $u_1$, $u_2$, $v_1$, and $v_2$ are compared for each horizontal direction in the analysis area, based on the pressure. Figures 20 and 21 show the results of the comparison of the values of the pressure and viscosity coefficient, and Figures 22 and 23 show the results of the comparison of the values of the pressure and flow velocity. The values of the pressure and the viscosity coefficient indicate that the values of the viscosity coefficient are distributed irrespective of changes in the pressure. From Equation (4), it is considered that the value of the viscosity coefficient is a parameter that is not determined only by the pressure. It was found, from this, that the viscosity coefficient is adjusted to almost the same value in the downstream part by combining the effects of the shear strain rate and pressure.

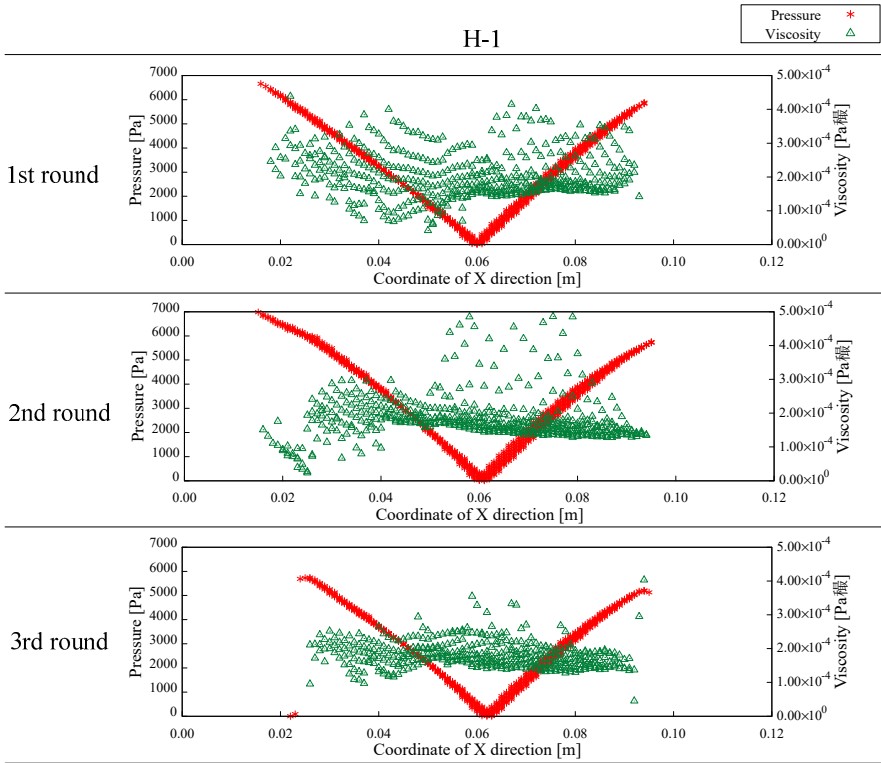

**Figure 20.** Comparison of the values of the pressure and viscosity coefficient from the simulation results of H-1.

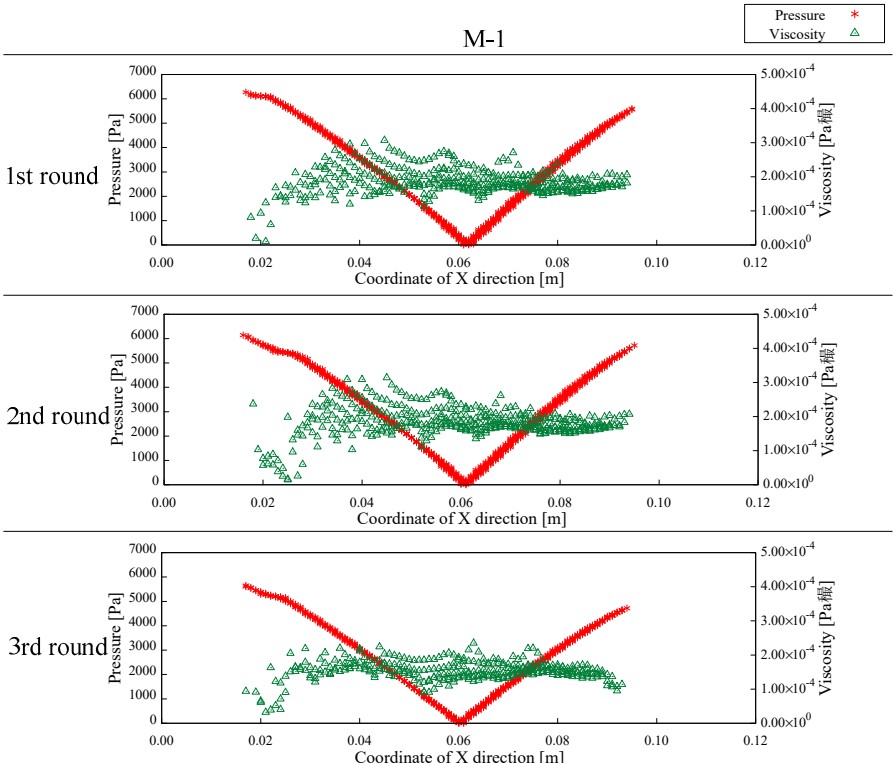

**Figure 21.** Comparison of the values of the pressure and viscosity coefficient from the simulation results of M-1.

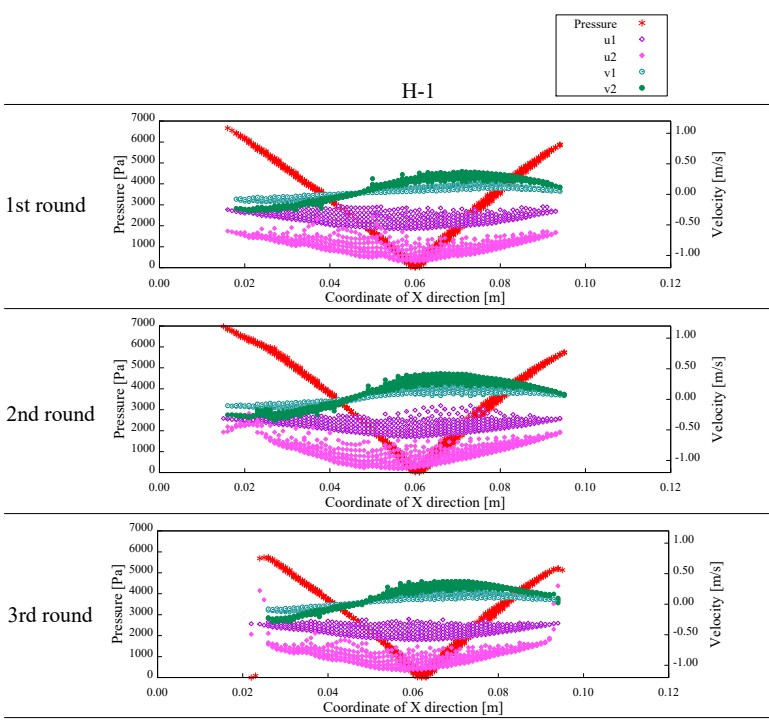

**Figure 22.** Comparison of the values of the pressure and flow velocity from the simulation results of H-1.

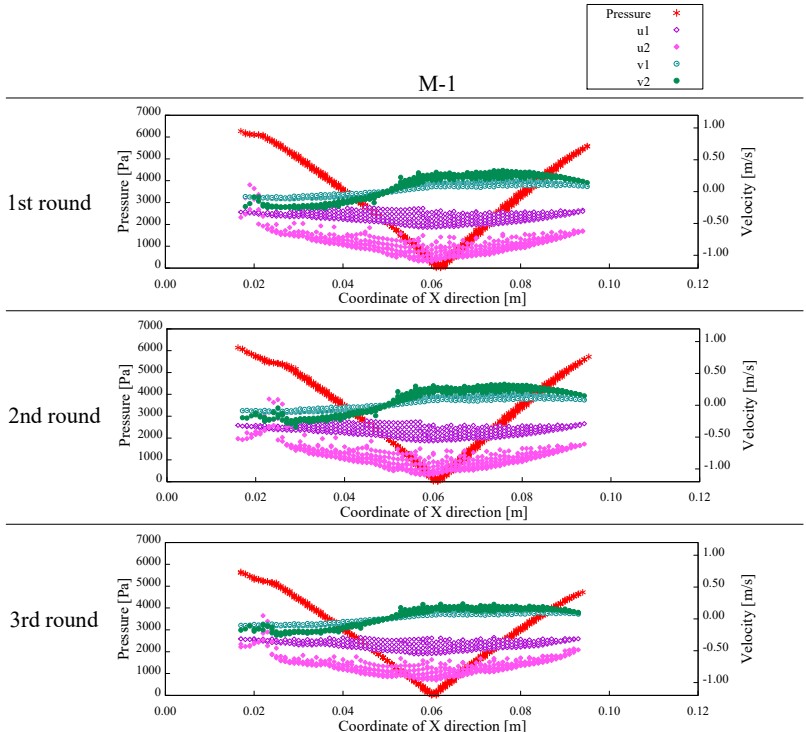

**Figure 23.** Comparison of the values of the pressure and flow velocity from the simulation results of M-1.

On the other hand, from the values of the pressure and flow velocity $u_1$, $u_2$, $v_1$, and $v_2$, the vertical flow velocity changes in a similar way to the pressure value, and the horizontal flow velocity changes at around the position where the minimum pressure value occurs. This indicates that the acceleration

of the pressure decreases from the beginning of the flow to the center of the lower part, and then it increases again due to a change in the flow with deceleration. In other words, it is considered that the flow behavior of the object changes at around the position where the pressure reaches the minimum. Therefore, we focused on 0.07 m to 0.08 m as the acceleration range, and 0.04 to 0.05 m as the deceleration range. In addition, we discussed the relationship between the shear strain rate and the viscosity coefficient and the relationship between the shear strain rate and the internal friction angle.

Figure 24 shows the relationship between the shear strain rate and the viscosity coefficient. It is found, from this result, that the viscosity coefficient in the acceleration region tends to increase from a shear strain rate of around 80.0 s$^{-1}$ and gradually increases. Moreover, in the deceleration region, the variation of the viscosity coefficient decreases from a shear strain rate of around 100 s$^{-1}$ and tends to converge at about $2.00 \times 10^{-4}$ Pa·s. For the above-mentioned results, it is considered that the flow of the artificial avalanche used in this experiment show dilatancy in the acceleration region and plasticity in the deceleration region. It has been reported that dilatancy is possibly exhibited on real avalanche flow, e.g., Bartelt [28]. Therefore, it is considered that the results obtained in this experiment are not abnormal.

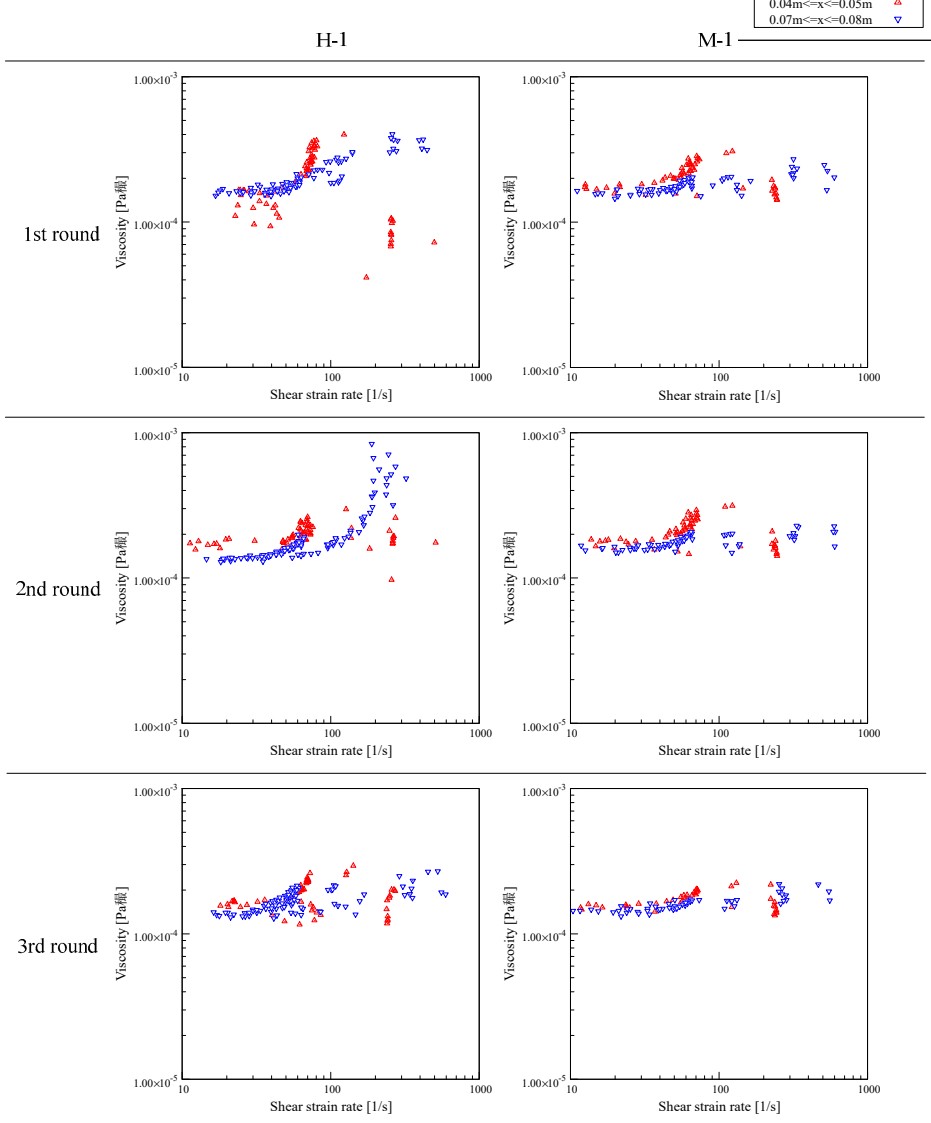

**Figure 24.** Relationship between the shear strain rate and the viscosity coefficient from the simulation results. The red mark shows 0.04 to 0.05 m as the deceleration range, and the blue mark shows 0.07 m to 0.08 m as the acceleration region.

Figure 25 shows the relationship between the shear strain rate and the internal friction angle. From these results, it can be confirmed that the internal friction angle increases as the shear strain rate increases. In this study, the snow sample condition is dry and the temperature is kept constant. Therefore, it is presumed that the cause of these results is from the effects of the snow sample condition. In addition, the variation of the internal friction angle tends to be lower in the region where the shear strain rate is low and higher in the region where the shear strain rate is high. This tendency is the same with respect to the acceleration and deceleration ranges. From the above-mentioned results, it is suggested that the variation of the internal friction angle depends on the shear strain rate. These results suggest that the frictional changes inside the avalanche are strongly influenced by the surge, which is similar to that reported by Köhler et al. [27].

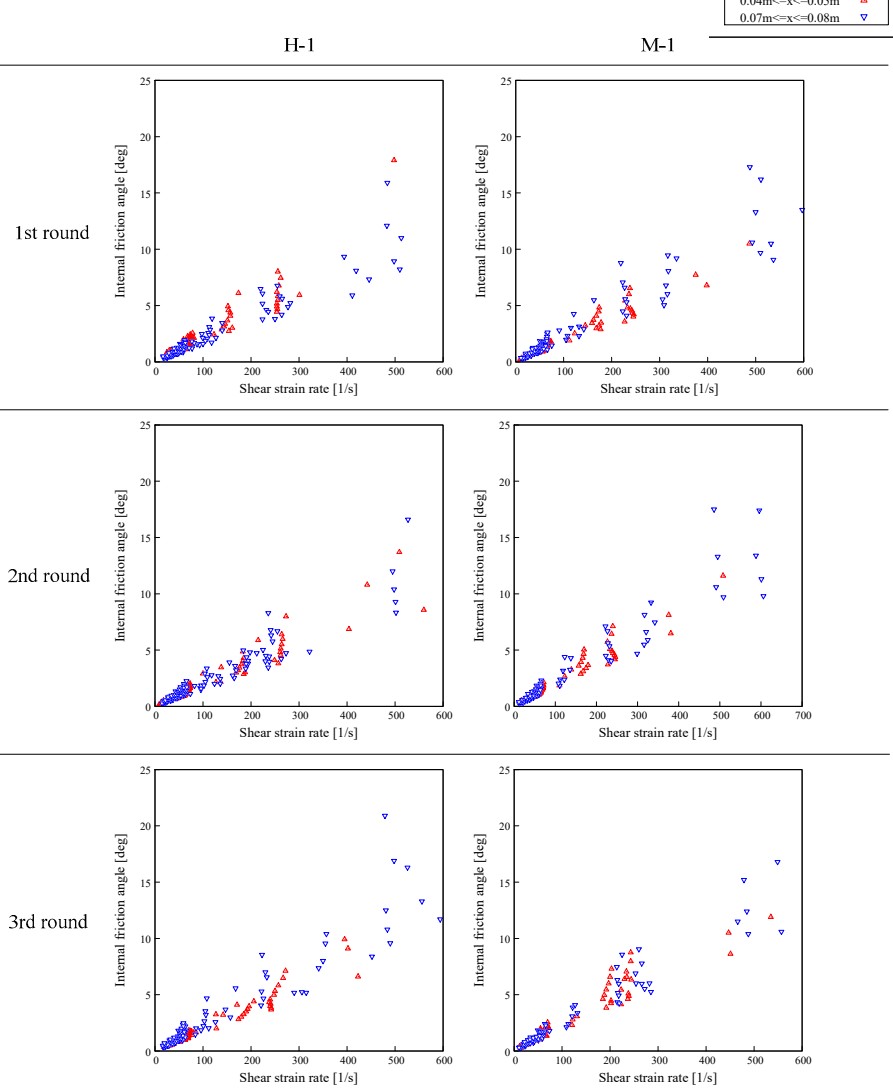

**Figure 25.** Relationship between the shear strain rate and the internal friction angle. The red mark shows 0.04 m to 0.05 m, as the deceleration range, and the blue mark shows 0.07 to 0.08 m, as the acceleration region.

The results obtained by this study are based on the trends of viscosity coefficients by simulations, and it is not clear whether the results can be applied to real avalanches or not. Then, the viscosity coefficient obtained from the simulation result is substituted for the denominator of Equation (23), which obtained the Reynolds number of the non-Newtonian fluid, and the Reynolds number of the pseudo-flow is calculated again using the same coefficients as in Figures 13 and 14 for the molecular

components. The results of the obtained Reynolds number are shown in Figure 26a. These results show that the Reynolds number using the viscosity coefficients obtained in the inverse simulation was calculated more than 1000 times larger than the values in Figures 11 and 12. Although we cannot assert that the Reynolds number obtained in Equation (23) perfectly represents the non-Newtonian viscosity, it is unlikely that the scales of the values obtained are at least significantly different. These differences may be attributed to the excessively small viscosity coefficients obtained by the numerical simulation. Therefore, accurate viscosity coefficients may not be obtained at the scale and density of the model experiments used in this study. In order to check the influence of scale and density, we consider the results obtained by assuming a scaled-up model based on the Froude number obtained by Equation (22). If the layer thickness of the pseudo-flow used for the representative length of the model experiment is 10 times or 100 times, the mesh size used in the simulation also increases proportionally. Therefore, the mesh size is 10.0 mm and 100 mm, and the flow velocity is also $\sqrt{L_m/L}$ times, which is about 3.16 times and 10.0 times. The density of the snow was the same as that used in the experiment. The Froude number at the time of scaling up is the same for all simulations, and only the size of length is scaled up with no change in the flow rate. First of all, the Reynolds number obtained from the simulation is shown in Figure 26b,c. It should be noted that the Reynolds numbers shown in Figure 15 do not change when using the representative values when scaled up. The results using the viscosity coefficients obtained by simulations are similar to those shown in Figure 15 as the mesh size increases, while the values are almost the same for a 100 times larger mesh size.

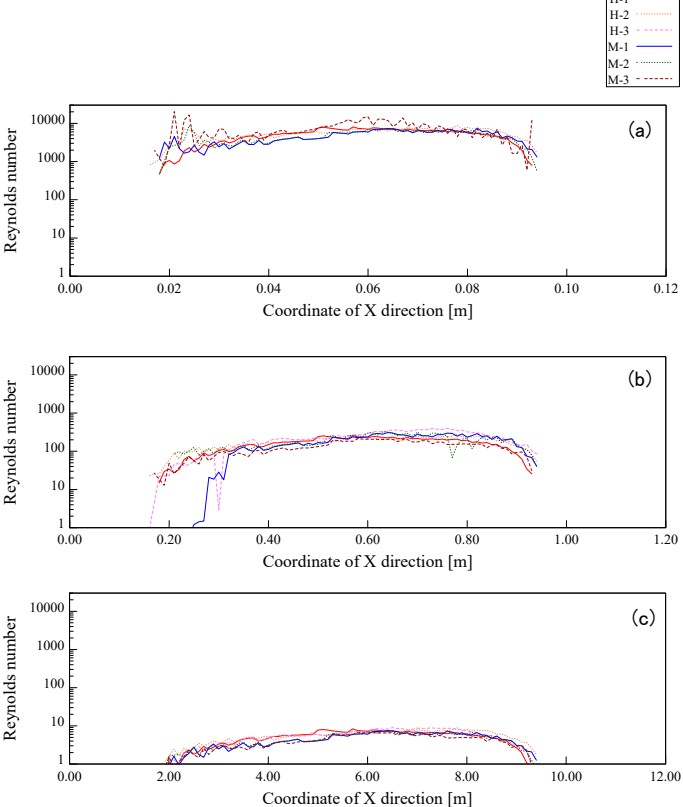

**Figure 26.** Relationship of Reynolds numbers in each cross section using the viscosity coefficient obtained from the simulation result: (**a**) the mesh size is 1mm; (**b**) the mesh size is 10 times; (**c**) the mesh size is 100 times.

Next, the relationship between the shear strain rate and the viscosity coefficient is shown in the Figure 27. The viscosity coefficients obtained by simulations increase with scaling up, and the trend is similar for all results. The pseudo-flow at a mesh size of 100 times larger than the real avalanche is about 12.5 m in length and 0.50 m in thickness, and the use of 300 kg·m$^{-3}$ snow density, which is close

to the real avalanche, for 1/100 scale model experiments may be a factor to break the similarity law. However, the trends in viscosity coefficients obtained numerically are similar, even for different density results. It is difficult to obtain accurate viscous properties using small model experiments, since it has been reported that real avalanches may scale up in turbulence and instability as the structure of the flow becomes more complex as the scale increases [27], but if the viscous properties show a pronounced shear strain rate dependence, it is possible to get a rough trend.

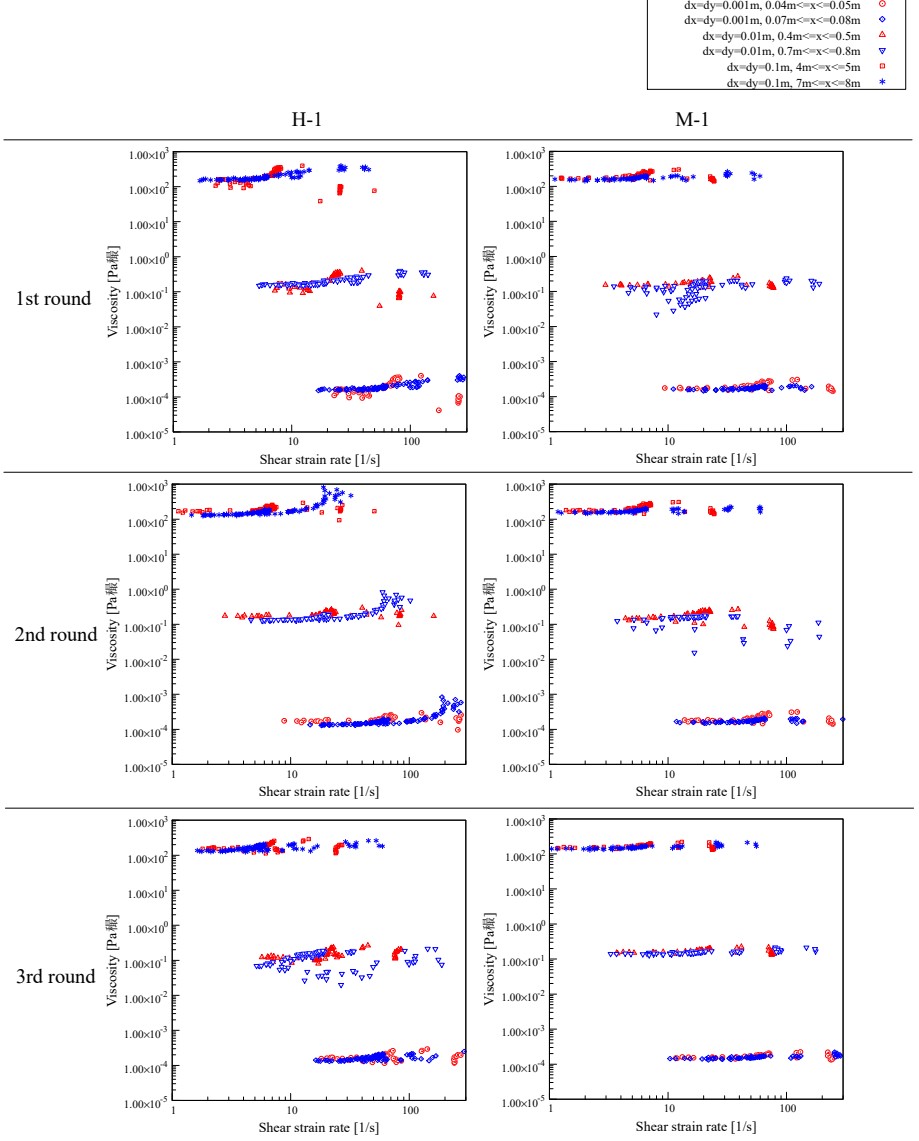

**Figure 27.** Relationship between the shear strain rate and the viscosity coefficient from the simulation results. The red mark shows 0.04 to 0.05 m as the deceleration range, and the blue mark shows 0.07 m to 0.08 m as the acceleration region.

It is concluded that the proposed method for estimating flow and friction properties from the pseudo-flow generated by the rotating drum using the time inversion analysis has a possibility to analyze the viscous properties of the model experiment only for the flow in which the shear strain rate dependence is pronounced, although the scale of the model experiment and the density of the material used remain to be adjusted precisely.

In addition, since the application to real avalanches needs to be analyzed accurately, it is expected that the application will be expanded by scaling up the model experiment using a rotating drum or by

performing the same study with velocity distributions extracted from observations using a full-scale avalanche chute.

## 6. Conclusions

In this study, in order to understand the flow characteristics of an artificial avalanche, we proposed a flow observation method using a rotating drum device and the flow characteristics extraction method using inverse simulation. The following results were obtained:

- It became possible to continuously observe the flow of artificial avalanches using the rotating drum device.
- The time inversion simulation using the observed flow velocity of the pseudo-flow as an input parameter shows that the method of qualitatively extracting the flow characteristics of a flowing part may be used only for flows where the shear strain rate dependence is prominent.
- In the flow characteristics of the pseudo-flow divided into accelerated and decelerated ranges, it is confirmed that the flow characteristics of the snow particles are plastic but include non-Bingham properties, and the internal friction angle is also obtained as a parameter that depends on the shear strain rate. However, it was found that the application to the real scale remains to be a challenge.

Moreover, the method proposed in this study did not successfully extract the flow condition of the artificial avalanche when the snow particles were 2 and 4 mm. This could be due to the fact that the movement of snow particles could not be tracked in the mesh, because the set mesh size of the simulation in this study was 1 mm. On the other hand, the thickness of the artificial avalanche was about 10 mm. Therefore, the numerical error included in the analysis results probably increased when the analysis mesh size was 2 mm or more.

In the future, the dependence of the analytical mesh size should also be clarified when scaling up the model experiments for application to real avalanches. In addition, although dry snow particles were used in this study, it is necessary to perform the same kind of verification for wet snow conditions. Therefore, it is important to consider situations in which the temperature can be controlled and the properties are less likely to change, even in wet snow conditions.

**Author Contributions:** Conceptualization, K.O., K.N., Y.K. and J.-i.S.; methodology, K.O., K.N., Y.K. and J.-i.S.; software, K.O.; validation, K.O. and K.N.; formal analysis, K.O.; investigation, K.O.; data curation, K.O. and K.N.; writing—original draft preparation, K.O.; writing—review and editing, K.O., N.K., Y.K. and J.-i.S.; visualization, K.O.; supervision, Y.K. and J.-i.S. All authors have read and agreed to the published version of the manuscript.

**Funding:** This research received no external funding.

**Conflicts of Interest:** The authors declare no conflict of interest.

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
