# Peer review of "Inverse Simulation for Extracting the Flow Characteristics of Artificial Snow Avalanches Based on Computation Fluid Dynamics"

_geosciences, doi:10.3390/geosciences10060221_

Round 1

Reviewer 1 Report

I would very much like to see this paper published.  However, you do not seem to be aware of on-going research and results in the snow avalanche field.  For example, numerical simulation programs have been used since 2000(!) in Europe -- furthermore, there is much literature concerning "dilatancy" in avalanches.  I think the introduction should therefore be rewritten to accurately depict the main questions in the field.

Reviewer 2 Report

This is a potentially interesting paper on the variability of a viscosity coefficient in small-scale drum-based avalanches. However, this statement contains a number of issues in its own terms and relative to the field scale avalanches that the results are meant to be relevant for:
(a) Is a Bingham model appropriate for a flowing avalanche, or is the notion of viscosity ill-posed?
(b) How relevant are small-scale experiments at low velocity for the dynamics of actual avalanches?
(c) Variation in friction parameters has been known about for a long time in avalanche models but this can result from erosion (Sovilla et al.), or roll-wave instabilities (the pioneering measurements by Vriend et al., 2013 in Geophysical Research Letters). Are these processes occurring in the experiment correctly at low velocity? If not how relevant are the experiments.
None of these questions are answered or really discussed in the paper. I've provided more comment on these below.

MAJOR POINTS
I'm not sure that the meaning of the opening sentence of the introduction is clear. Numerical models have been used for practical snow avalanche work for some time. For example, work by Voellmy (1955) long-underpinned applied avalance modelling in Switzerland. Perla et al. (1980) underpinned work in Canada, and Norem et al. (1986) work in Norway. Margarita Eglit wrote a review paper about 10 years ago in Cold Regions Science and Technology describing Russian work from the 1960s. In terms of more fluid dynamics types of simulations, Barbolini et al. (2000) compared and contrasted a variety of such models. In contrast the authors say that this has emerged recently and only cite papers since 2017. As such, the wealth of past work is under-represented. Eckert et al. (2012) reviewed applied avalanche risk analyses in Cold Regions Science and Technology. They covered modelling the dynamics as part of that work.

A key statement to underpin the rationale for the paper is on L37 where it is stated: "Many numerical analysis methods for predicting the motion of snow avalanches set the characteristics of the static friction obtained from mechanical tests, such as the triaxial compression test and the direct shear test." If there are "many" such studies, the authors should reference some of them at this point rather than further down the page. Quite often, the parameterization for applied models is done retrospectively, not from a physically-based value for static friction because of the difference between that value and the values needed to get the avalanche to flow far enough. These two philosophies need to both be articulated, with references to literature adopting both. This is particularly important for this paper because, while an attempt is made to provide a physical basis for the friction parameters, the approach taken rather undermines this by assuming that the snow is incompressible. Hence, the friction parameters will still need calibration in the field after this study because such effects are not being incorporated. Therefore, at minimum, the calibration philosophy should be articulated.

It is not clear how relevant the experiment is to the dynamical conditions in an avalanche. At the very least, the Froude number needs to be stated so that it is clear that the experiment is providing data that are relevance to the dynamical regimes seen in the field. If this is not of dynamical relevance for the field situation, then it is not at all clear that the rationale for the paper is correct, which would question the merit of publication.

Figures 12-16. The minimum font size on a figure should be about 8 pt. Some of these figures contain important text at about 2 pt and all contain text that is far too small. Figures need changing so that the scientific information contained in these figures can be studied effectively. I really could not read them.

MINOR POINTS
Line 60. Nicolas Eckert has just published a paper (J. Glaciology, 2020) which obtains dynamic friction information from photogrammetry data. This paper should be cited as it provides an example of how this may be done.
Line 179 - SMAC should be defined and a reference provided.
L249 - "The airflow was incorporated". At a high velocity in an avalanche, one would expect a fluidising upper surface of enhanced permeability that acts as a source of drag. To model this effect would require a genuine two-phase computation that has not been done. Thus, the authors need to be more specific about what it is that they have incorporated in terms of air flow dynamics.
Reference 7 - the first letters in each word of the journal title needs capitalisation.
Reference 11 - segregation not seregation

Round 2

Reviewer 1 Report

This is the second version of the paper. 

The introduction has been rewritten to stress the role of wet snow avalanches in Japan, and the problems of modelling wet snow avalanches in general, especially "wet snow" friction.  I believe this section is improved, however, sets the paper in a little bit different direction then the first version.

I think a very interesting paper is from Jomelli and Bertran (2001) entitled "Wet snow avalanche deposits in the French Alps" which discusses the properties of wet snow avalanches.  Another interesting paper is "Granulometric investigations of snow avalanches" by Bartelt and McArdell where important differences between wet snow granules and dry snow granules are discussed.

Perhaps the only paper that I know that deals with the actual modelling of wet snow avalanches is the paper by Vera Valero et al. that appeared in 2016 entitled "Modelling wet snow avalanche runout to assess runout safety at a high-altitude mine in the central Andes", Nat. Hazards Earth Science Systems, 16, 2303-2323.

Wet snow avalanches certainly involve the simulation of temperature and frictional melting.  A paper discussing the simulation of avalanche temperature is contained in another paper by Mr. Vera Valero, "Release temperature, snow-cover entrainment and the thermal flow regime of snow avalanches", Journal of Glaciology, Volume 61 (225), 2015.

Perhaps the authors will find information to support their ideas in the conclusions.  The depiction in the introduction that wet snow avalanches involve only the selection of parameters is not quite true.  In fact much work has been done to try to find wet snow models.  This reviewer was not aware of the stress placed on wet snow avalanches, until reading the second reviewer.  Clearly, somewhere in the conclusions the authors should stress the importance of gathering real-scale field data.

I presume that a second reviewer asks the authors to perform a scaling using Fr. and Re nos.  These dimensionless numbers unfortunately do not account for "wetness" or some measure of the dilatancy.  Especially the important properties of wet snow.  Therefore, I personally do not believe that the comparison is useful or inciteful because these numbers do not capture the complicated physics behind wet snow avalanches.  I think the authors are aware of this, but are just trying to accomodate the wishes of the reviewers. 

Reviewer 2 Report

In some ways I am less convinced about this manuscript than I was the first time around. When you ask the authors to do something and they do not do so, or do it in a very strange way, it raises questions in the reviewer's mind.

In terms of specifics, I asked the authors if the Bingham rheology and the use of a viscosity was ill-posed. They just commented that they are only interested in wet avalanches. But in which case, the title of the paper needs changing as it does not state this and nor does the abstract or conclusion. Of course, stating that one is concerned with wet snow avalanches does not provide a justification for the use of a Bingham rheology. Lines 161-162 simply state that the Bingham model is versatile and a soils paper is cited. It may be versatile, but it is right? To justify it, the authors need to deal with the work looking at snow rheology that comes to very different conclusions and argue why this science is not relevant:

Kern, Tiefenbacher and McElwaine, Cold Regions Science and Technology (2004) found a Herschel-Bulkley rheology was most appropriate in their experiments on flowing snow. Sovilla et al. (2010) looked at avalanche deposits and suggest a Pouliquen model or a cohesive friction model for the tail of the avalanche (which will be more similar to a wet avalanche). The authors stated in their rebuttal that the roll-wave instability noted by Vriend et al. (2013) has not been verified. That is not the case. Kohler et al. (2016) in JGR also found such instabilities and I quote from their abstract, "The small or “minor” surges appear to be a roll wave-like instability, and these can greatly influence the front dynamics as they can repeatedly overtake the leading edge. We analyzed the friction acting on the fronts of minor surges using a Voellmy-like, simple one-dimensional model with frictional resistance and velocity-squared drag. This model fits the data of the overall velocity, but it cannot capture the dynamics and especially the slowing of the minor surges, which requires dramatically varying effective friction." None of the references on lines 62-72 provide validation data for this rheology and a biviscous Bingham model was used in the early 1980s before it was rejected. Hence, the authors need to make a scientific case to justify their rheology. If they cannot, the paper must be rejected. 

I asked the authors to state their Froude number to show that their experiments are relevant to actual snow avalanches. The new addition on L253-269 never actually states the Froude number and inserts a particle diameter rather than a flow depth in to the denominator. The authors also state that the "the maximum average flow velocity at the center of the lower stream is about 1 m/s", but Froude numbers are not based on maximum velocities, they are based on bulk velocities. If one estimates the average velocity from Fig. 7 and 8 it is about 0.1 m/s for one and 0.3 m/s for the other. Based on the authors' stated depth of 1 cm, this gives a Froude number of 0.3 to 1, i.e. sub-critical. Hence, it is not at all clear that the experiments are dynamically relevant for very many avalanches. If publication is to proceed, the authors must make it much more clear in the abstract, introduction and then throughout as to exactly which avalanches they are trying to simulate. It would appear that they are interested in events that are slow moving and associated with spring-melt rather than the events most people try to model that move rapidly and cause risk to towns. As such, the title needs changing as the authors seem to only be interested in a class of slow moving wet avalanches.

On lines 177-178 the authors state "Generally, snow is compressible material. However, in this study, snow is assumed to be an incompressible material." All the technical detail on p.7 - 8 is therefore, based on the authors' own statement, irrelevant for avalanches. How can the authors justify assuming incompressibility when they know the snow is compressible, given their chosen Bingham model (i.e. returning to the first point from a different angle)? This will potentially have a massive impact on their calculated friction coefficient, particularly given the low Froude number implies the dynamical regime will be very much about internal yielding within a cohesive mass (rather than granular collisions). The authors need to use their experimental data to demonstrate the degree of error that follows from this assumption because as things stand, L177-178 and the subsequent work implies that the authors obtain a friction coefficient that applies to hypothetical, incompressible snow, but the rationale for the study as stated in the introduction and abstract is to produce more realistic coefficients for real avalanches.

It is good that I can now see the information in figures 15-20 although they all still contain important mistakes (see below). However, it is not clear how the errors in the experimental post-processing are used in the inverse model. Presumably the best simulation is one that minimises the variability in the friction parameter by using confidence intervals on the velocity estimates in the experiment (to avoid over-fitting). This should be discussed.

In eq. 22 the nature of the characteristic length should be stated. This is presumably flow depth?

The y-axes in Fig. 15-20 need some work. There is no label on the left-hand axis and the units for the right-hand axis (in Fig. 15-18, left-hand axis in Fig. 19 & 20) are not formatted properly. This is particularly unfortunate for Fig. 17 & 18 as the right-hand y-axis label is directly on top of the legend making this figure still very difficult to read.

The authors must use superscripts properly when typesetting standard form and ensure that the "\times 10" part is included rather than writing "e-4" or similar.

Round 3

Reviewer 2 Report

This new version is a significant improvement on what has gone before. A much better introduction leads in to a more critical discussion of the rheological assumptions. The recognition that the Froude number is only just supercritical is now explicit, meaning that the reader can much more readily understand the domain of application of the experiments undertaken.

There are some minor issues with English that can hopefully be dealt with during the editorial/publication process, but otherwise, I am happy to deem this acceptable.